Detection of cyberhate speech towards female sport in the Arabic Xsphere

Alhayan Fatimah fnalhayan@pnu.edu.sa
Almobarak Monerah
Shalabi Hawazen
Alshubaili Luluwah
Albatati Renad
Alqahtani Wafa
Alhaidari Nofe
Department of Information Systems, College of Computer and Information Sciences, Princess Nourah bint Abdulrahman University , Riyadh , Saudi Arabia
Vassileva Julita
Electronic publication date: 2024 Jun 27
Publication date: 2024
Volume: 10
Electronic Location ID: e2138
Received 2024 Jan 25; Accepted 2024 May 28
Copyright: ©2024 Alhayan et al.
Copyright year: 2024
Copyright holder: Alhayan et al.
License: This is an open access article distributed under the terms of the Creative Commons Attribution License, which permits unrestricted use, distribution, reproduction and adaptation in any medium and for any purpose provided that it is properly attributed. For attribution, the original author(s), title, publication source (PeerJ Computer Science) and either DOI or URL of the article must be cited.
License URL: https://creativecommons.org/licenses/by/4.0/

Keywords: Natural language processing, Hatespeech, Female sport, Machine learning

Funding: Princess Nourah bint Abdulrahman University Researchers Supporting Project PNURSP2024R719 Princess Nourah bint Abdulrahman University, Riyadh, Saudi Arabia This work was supported by the Princess Nourah bint Abdulrahman University Researchers Supporting Project number (PNURSP2024R719), Princess Nourah bint Abdulrahman University, Riyadh, Saudi Arabia. The funders had no role in study design, data collection and analysis, decision to publish, or preparation of the manuscript.

==============================
The recent rapid growth in the number of Saudi female athletes and sports enthusiasts’ presence on social media has exposed them to gender-hate speech and discrimination. Hate speech, a harmful worldwide phenomenon, can have severe consequences. Its prevalence in sports has surged alongside the growing influence of social media, with X serving as a prominent platform for the expression of hate speech and discriminatory comments, often targeting women in sports. This research combines two studies that explores online hate speech and gender biases in the context of sports, proposing an automated solution for detecting hate speech targeting women in sports on platforms like X, with a particular focus on Arabic, a challenging domain with limited prior research. In Study 1, semi-structured interviews with 33 Saudi female athletes and sports fans revealed common forms of hate speech, including gender-based derogatory comments, misogyny, and appearance-related discrimination. Building upon the foundations laid by Study 1, Study 2 addresses the pressing need for effective interventions to combat hate speech against women in sports on social media by evaluating machine learning (ML) models for identifying hate speech targeting women in sports in Arabic. A dataset of 7,487 Arabic tweets was collected, annotated, and pre-processed. Term frequency-inverse document frequency (TF-IDF) and part-of-speech (POS) feature extraction techniques were used, and various ML algorithms were trained Random Forest consistently outperformed, achieving accuracy (85% and 84% using TF-IDF and POS, respectively) compared to other methods, demonstrating the effectiveness of both feature sets in identifying Arabic hate speech. The research contribution advances the understanding of online hate targeting Arabic women in sports by identifying various forms of such hate. The systematic creation of a meticulously annotated Arabic hate speech dataset, specifically focused on women’s sports, enhances the dataset’s reliability and provides valuable insights for future research in countering hate speech against women in sports. This dataset forms a strong foundation for developing effective strategies to address online hate within the unique context of women’s sports. The research findings contribute to the ongoing efforts to combat hate speech against women in sports on social media, aligning with the objectives of Saudi Arabia’s Vision 2030 and recognizing the significance of female participation in sports.

Introduction

Social networking sites like X (formerly known as Twitter) and Facebook provide rapid connection and interaction with diverse individuals, facilitated by advancements in technologies such as high-speed internet and portable devices. The exponential increase in users over recent years has made social media an integral part of modern life (Ausat, 2023), fostering unprecedented social connectivity. While the accessibility of cyberspace empowers users with autonomy and the free exchange of ideas, it also heightens exposure to detrimental content such as online bullying and cyberhate speech.

Gendered hate speech and its consequences

Hate speech is defined as spreading hatred, violence, and discrimination based on protected characteristics, such as race, ethnicity, gender, religion, and others (Kilvington, 2021). Research indicates that hate speech has dire consequences, leading to crimes and profound psychological issues (Chaudhary, Saxena & Meng, 2021). These harms are observed at individual, collective, and societal levels. On the individual level, hate speech can lead to poor performance at work or school and harmful psychological effects, such as sadness, anxiety, insomnia, drug abuse, self-harm and even suicidal ideation. Collective harms affect a group of individuals by lowering the quality of the online community and making people emotionally insensitive to hateful statements. Societal harms include group division, collective trauma, and hate crime violence, requiring security, healthcare, and legal considerations (Chaudhary, Saxena & Meng, 2021).

Gendered hate speech specifically attacks individuals based on perceived gender or sex, primarily targeting women and girls. This form of hate speech, prevalent on social networks, aims to degrade, humiliate, and marginalize women (Chetty & Alathur, 2018). The Council of Europe’s European Commission against Racism and Intolerance (ECRI), an independent human rights watchdog, defines gendered hate speech as any expression aimed at expressing contempt towards a person based on their sex or gender, or reducing them to their sexual dimension (Council of Europe Gender Equality Strategy , 2016).

Social networks are the main platform for gender-based online harassment (Simons, 2014). Accepting gendered hate speech is often expected of women, yet it inflicts severe psychological, emotional, and physical harm. This form of hate speech aims to degrade or humiliate them, disparage their abilities and beliefs, damage their reputation, make them feel helpless and subservient to men, and penalize them for not acting in a certain way. It is thus crucial to study online hate speech targeting women (Mohammed, Abdullah & Hussein, 2022). This study begins to fill that gap by focusing on online hate speech in the sport domain, targeting Arabic women sport in particular. This is especially pertinent given the significant contributions of sports to overall well-being, including physical health and emotional happiness, as recognized in recent research (Vveinhardt, Bite Fominiene & Andriukaitiene, 2019). Encouraging an active lifestyle not only combats non-communicable diseases like obesity and diabetes but also enhances quality of life (Al-Shahrani, 2020). Over the past decade, there has been growing recognition of the importance of sport for personal and social development (Vveinhardt, Bite Fominiene & Andriukaitiene, 2019). Recognizing its significance, sport is a strategic element in Saudi Arabia’s Vision 2030, aiming to integrate it into societal culture, raise public awareness of its health benefits, and support its economic role (Fakehy et al., 2021). However, Arab women’s participation in sports remains low due to various sociocultural and health-related factors (Almudaires, 2022). Societal factors include restrictive perspectives on female athletes, with women’s participation in sports only being permitted since 2012, leading to lingering suspicions. Health issues, such as physical disabilities and specific conditions, further hinder participation. Of these, sociocultural barriers, driven by a discriminatory macho culture, are most significant, discouraging women from overcoming challenges and engaging in sports. Almudaires (2022) showed that very few Saudi women participate in sporting activities and events, only 55.6% of the 618 Saudi women surveyed, and the barriers rated by them were 75.7% on average, which include the media and social media that discourage female participation in sports, as well as negative cultural views due to conservatism and shame. In Saudi Arabia, low participation rates in sports among females and adolescent males significantly impact the sports industry’s performance, with broader implications for the country (Fakehy et al., 2021). To address this issue, scientific strategies aligned with Saudi Arabia’s intellectual renaissance are necessary to raise public awareness about the benefits of sports and physical activity. By promoting sports participation across all demographics, regardless of gender, benefit the economy by reducing healthcare costs, absenteeism, and sick days while boosting worker productivity (Fakehy et al., 2021).

Problem statement

Social media increasingly encourages darker behaviors and serves as a platform for abuse and discrimination (Kavanagh, Jones & Sheppard-Marks, 2016), including gender-based cyberhate targeting women’s participation in sports (Kavanagh, Litchfield & Osborne, 2019; McCarthy, 2022).

While hate speech detection in English has been extensively studied, its detection in Arabic content remains underdeveloped (Abuzayed & Elsayed, 2020). Studies in Arabic primarily focus on general detection or specific domains like religion or politics. To the best of our knowledge, research on cyberhate targeting Arabic female athletes, fans, and women’s sports on social media is lacking. Moreover, identifying hate speech in Arabic is particularly challenging due to the language’s rich and complex nature and structure, the variety of dialects, writing from right to left, neglect of diacritics (Alshutayri & Atwell, 2018). Researchers often rely on region-specific data and specialized algorithms to address these challenges (Al-Hassan & Al-Dossari, 2022).

This gap offers an opportunity for a thorough investigation into the prevalence and forms of online hate speech targeting this demographic, revealing the unique challenges faced by Arabic-speaking female athletes in the digital landscape. The aim is to propose an automated solution for detecting Arabic hate speech against women’s sports, focusing on X, the prominent social networking platform widely used for discussing sports in Arabic.

Purpose of study

The aim of this research is to propose an automated solution for detecting Arabic hate speech against women’s sport, focusing on X, the most widely used social networking platform to exchange ideas and opinions on sport in Saudi Arabia.

The contributions of this study are threefold:

(1) It identifies the different forms of online hate aimed at Arabic women in sports.

(2) It constructs a hate speech dataset in Arabic, specifically pertaining to women’s sports. This dataset contains 7,487 tweets and it has been manually annotated by six annotators. In this dataset, 2,025 tweets contain hate, 2,025 do not contain hate, and 3,675 are irrelevant to the topic at hand. This dataset is a valuable resource for further research aim at combating hate speech targeting Arabic women in sports.

(3) It develops machine learning classification models to identify hatful tweets pertaining to women’s sports, using commonly used feature sets in natural language processing (NLP) tasks.

In the next section, a summary of related work is provided. In ‘Materials and Methods’ and ‘Discussion’, the data collection for and results of Study 1 and study 2 are presented. ‘Limitations and Future Research Directions’ contains the discussion, followed by the conclusions in ‘Conclusions’.

Related Work

Online hate speech and sport

Online hate speech has grown exponentially worse with the development of social media. Online hate speech in the context of sports is any hostile or discriminating comments or actions directed at athletes, teams, or supporters on the internet. This can manifest itself in a variety of ways, including trolling, cyberbullying, ableism, sexism, and racist insults. Hate speech can be directed at particular athletes, teams, or even entire genders, and it can seriously harm the mental health of individuals who are the targets. Examples are the racist abuse English football players were subject to during Euro 2020 and the culture wars that have ignited over Colin Kapernick’s taking the knee in protest against systemic racism in America. This has brought increased attention to the topic of online hate in sport (Blanco-Castilla, Fernández-Torres & Cano-Galindo, 2022).

The issue of online hate speech in sports, especially towards female sports athletes and journalists, is a growing concern. Kearns et al. (2022) provide a scoping overview of the literature on online hatred in sports, noting that there is a need for additional investigation into this pressing subject. The review emphasises the importance of involving diverse stakeholders, including athletes, journalists, policymakers, and social media platforms, to address this pervasive issue. The authors also emphasise the importance of paying closer attention to online hate speech towards women in journalism and sports. It is critical that all parties involved work to address this widespread problem and advance a welcoming and safe online space (Kearns et al., 2022).

The study by Krieger et al. (2022) examined the differences between media coverage and Instagram conversations about women athletes. Researchers analysed comments left by men on posts featuring eight women athletes in ESPN’s “Body Issue” (Coyne, Santarossa & Dufour , 2022) and compared them with traditional media discourse. Results showed that online comments, like traditional media, focused on gender and athletic performance, with similar levels of sexism. The discussion emphasised the importance of intersectional perspectives in studying the harmful narratives that exist in online discourse about women athletes.

Kavanagh, Litchfield & Osborne (2019) shed light on the pervasive issue of online hate speech towards women in sports and journalism. Employing a netnographic approach and the lens of third-wave feminism, the researchers analysed social media criticism and fan interaction with regards to the top-five-seeded female tennis players during the Wimbledon Tennis Championships. The study highlighted that social media platforms have provided a breeding ground for gender-based cyberhate, particularly towards prominent women in the workplace, leading to self-censorship. The findings emphasise the need for greater attention, regulation, and accountability of online behaviour to promote a safe and inclusive online environment.

The issue of hate speech towards female sports journalists has been investigated in a study conducted by Blanco-Castilla, Fernández-Torres & Cano-Galindo (2022) that aims to establish the characteristics and magnitude of this problem and determine how it affects Spanish female sports journalists. In fact, 89.6% of Spanish female sports journalists who took part in this study reported that they have experienced hate speech and other types of harassment, both online and at work, with the main harassers criticising their ability to do their jobs or bringing up their appearance (Blanco-Castilla, Fernández-Torres & Cano-Galindo, 2022).

A study by Demir & Ayhan (2022) on online harassment experienced by female sports journalists in Turkey was not a random study, but rather a specific examination of the experiences of women in a particular field and location. It uses a qualitative approach to analyse the content of tweets directed towards female sports journalists, as well as interviews with journalists themselves. This methodology allows for a deeper understanding of the experiences and perspectives of female sports journalists in Turkey, but it also means that the findings may not be generalisable to other contexts or populations. Nonetheless, the study provides valuable insights into the challenges faced by women in sports journalism and the need for greater attention to issues of gender and inclusivity in the field.

These previous studies examined online hate speech related to sports in the context of English; however, there is a significant gap, as this this issue has not yet been studied in the Arabic context. This article thus provides an essential contribution, discussing the results of a qualitative study that investigated online hate speech targeting Arabic female athletes and fans.

Online hate speech detection

Studies discussed in the previous section highlight the need for urgent measures to address the problem of hate speech and other forms of online harassment towards women in sports. Hate speech in has historically been detected using classical machine learning, with emerging studies exploring the adoption of deep learning methods (Alshalan & Al-Khalifa, 2020). There are many studies about detecting hate speech using an English database. Burnap & Williams (2016) identifies cyberhate on X targeted towards individuals or social groups based on three divisions, race, sexual orientation, and disability. The machine classification experiment was performed using a support vector machine (SVM) algorithm with a linear kernel and a random forest decision tree algorithm using different combination of features, bag of words (BOW) features with hateful terms, and typed dependencies combined with BOW and hateful terms. The results of this experiment showed similar results for race and sexual orientation; in both experiments, the combination of BOW and hateful terms resulted in the highest precision rates. However, for disability, using hateful terms only led to all instances being classified as non-hate, and there was no improvement in using typed dependencies over BOW. That means that the cyberhate classifier would not maintain the same accuracy if the context changed (Burnap & Williams, 2016) .

Badjatiya et al. (2017) used a deep learning approach to detect hate speech on X. The aim of the study was to classify a text as sexist, racist or neither. The baseline method of this experiment used Char n-grams, TF-IDF (which uses the frequency of words to determine how relevant the words are) and bag of words. Convolutional neural network (CNN), long short-term memory (LSTM) and fast text were used, and they were all initialised with word embedding –either random embeddings or Global Vectors for Word Representation (GloVe) embeddings. The proposed method had better results on everything, precision, recall and F1 measure. Moreover, adding gradient boosting decision trees (GBDT) to the method led to better results, specifically when LTSM and random embedding and GBDT were all combined together (Badjatiya et al., 2017).

There are not a lot of studies that use Arabic datasets, so normally, when a researcher wants use an Arabic dataset, they need to gather the data themselves. Alshalan & Al-Khalifa (2020) uses a deep learning approach to detect Arabic hate speech of different types, such as religious, racist and ideologist, on X. The experiment was carried out using CNN, GRU, CNN+GRU, and BERT methods, in each case creating a binary classification in which tweets were classified as hateful or normal. Of the three neural models, the results show that CNN performed the best, as it had an F1 score of 0.79 and an AUROC score of 0.89. Compared to the other neural models, the BERT model failed to provide any improvement, which was unexpected, since BERT is known to be very powerful, achieving extraordinary results in NPL tasks such as sentiment analysis and question answering. However, this study’s focus was Arabic hate speech in general and not specifically that targeted at women or women in sports .

Al-Hassan & Al-Dossari (2022) classified Arabic hate speech tweets into five classes (none, religious, racial, sexism or general hate) using different machine learning algorithms. Support vector machine was used as a baseline model to provide a point of comparison against four deep learning machine languages, namely LTSM, CNN+LTSM, GRU, and CNN+GRU. Since the dataset was unbalanced, the two factors of comparison were recall and precision. The SVM model achieved 75% on the classifier, which suggests that it performed exceptionally well in distinguishing non-hate speech tweets, but unfortunately it did not perform well in identifying the different hate speech classes. The four deep learning machines outperformed the SVM model in the Arabic hate-speech multiclassification task. The CNN+LTSM Layer enhanced the overall performance of detection with 72% precision, 75% recall, and 73% F1 score.

Hate speech on social media targeted at women in sports is rapidly increasing (Kavanagh, Litchfield & Osborne, 2019; Kearns et al., 2022). Furthermore, it has recently been a controversial topic in Saudi Arabia (Al Ruwaili, 2020; Fakehy et al., 2021). While Almateg et al. (2022) have examined women’s sports in the Arabic language, as far as we are aware, no studies have specifically tackled the detection of gendered hate speech in Arabic sports discourse. Almateg et al. (2022) utilized sentiment analysis to identify the opinion of X users towards women’s sports in Saudi Arabia and to determine if there was a difference between X content before and after women’s sports became permitted in Saudi Arabia. Four hashtags were used to collect tweets, #Officially_female_sports_in_schools was used for the hashtag prior to women being allowed to partake in sports, #Women_Sport and #Female_Sport were used for both before and after, and #Tahani_Alqahtani (the name of a Saudi Arabian international-level athlete) was used for the hashtag after women had been allowed to participate in sports. This classification assigns labels to documents based on the number of words in two contrary lexicons, for example negative and positive sentiment. Tweets using the hashtag used before women’s sport was permitted in Saudi Arabia showed the need for women’s sports in Saudi Arabia, as long as it would be undertaken in accordance with Islam instructions. Tweets using the hashtag used after women’s sports had become permitted, #Tahani_alqahtani, were 72% in support of women participating in the Olympics. These results suggest that before women’s sports were permitted, people wanted women to be able to play sports, and afterwards they wanted women to be even more engaged in women’s sports.

Professional athletes are targets of online hate speech. Alsagheer et al. (2022) conducted a study focusing on three athletes that participated in the Tokyo 2020 Olympics, Simone Biles, Tahani Alqahtani and Laurel Hubbard. The dataset consists of 286 English hate speech comments collected from the social media platforms of the mentioned athletes, specifically YouTube, Facebook, and X, over four weeks. Active learning and supervised learning were used to analyse the data. Supervised learning can be deconstructed into three different approaches, Word2Vec, bag of words, and perspective results. The results of the experiments show Word2Vec obtains the best result in each subgroup in terms of accuracy, precision and F1 score; bag of words has the best result in active recall in all the supervised learning experiments; and perspective obtains the best result in total data accuracy, precision and F1 score. The results of the active learning experiment outperformed the supervised learning experiment, because it was able to alleviate the impact of skewed classes by selecting the most informative instances among unseen training data and minimised the possibility of wrong labels in training data by giving the domain expert a second chance to judge the data at run time.

Hate speech detection has attracted significant research interest recently, but the research has mostly been conducted using English datasets (Abuzayed & Elsayed, 2020). Moreover, there is a lack of available tools and resources to process the Arabic language compared to English. This is due to the complexity and richness of Arabic. Also, research use Arabic dataset from social media such as X had to not only to deal with the difficulties that come with using Arabic dataset, but also had to take into considerations X’s limitations. Tweets are usually short, and they are not always written with correct formal grammar and spelling. Often, there is borrowing of foreign words and using different words forms, and abbreviations are usually used to overcome the restricted lengths.

Based on the issues discussed above, the following research questions were formulated:

RQ1: What are the specific forms of hate speech experienced by female athletes and female sports fans on social media?

RQ2: How do the various machine learning models perform in identifying Arabic hateful tweets using linguistics features?

Materials & Methods

Study 1

Methods

Ethical approval was obtained from the Institutional Review Board of PNU University before commencing the study (23-0734). The data was collected through semi-structured interviews with The data was collected through semi-structured interviews with 33 Saudi females, who are either athletes, players for professional sports clubs, players for university clubs, or sports fans. Most of them mentioned having a presence on social media, either public or private. The majority of participants, totaling 22, are aged between 18 and 25. There are fewer participants, only 3, aged between 26 and 30, and just one person aged 31 to 36. Among the fans, 5 are aged between 18 and 25, while only 2 are 37 or older. This suggests that the majority of individuals involved are young, with only a few older participants and fans. Appendix S1A presents each participant’s demographics.

The interviews were conducted either face-to-face or online, based on the participant’s preference or request, demonstrating flexibility in the data collection process. The engagement with the participants commenced by presenting the study’s objectives and rationale, followed by obtaining their informed consent, ensuring ethical research practices. After the participants signed the consent forms, the interviews began.

The interview questions in Appendix S1B were crafted drawing from prior research on cyberhate and bullying, exemplified by studies like (Dredge, Gleeson & De La Piedad Garcia, 2014; Nyman & Provozin, 2019), with a focus of exploring the experiences of female athletes and sports fans with hate speech in online communities.

The average interview time was approximately 20 min. All interviews were audio-recorded. The researchers have transcribed all interviews and subsequently translated them into English.

Data analysis

The researchers analyzed the transcripts with directed content analysis, which “is the intellectual process of categorizing qualitative textual data into clusters of similar entities, or conceptual categories, to identify consistent patterns and relationships between variables or themes” (Julien, 2008).

Results

Based on the analysis, three themes were discovered that answer

RQ1: What are the specific forms of hate speech experienced by female athletes and female sports fans on social media?

Theme 1: Gender-based derogatory comments

Gender-based derogatory comments include derogatory comments and slurs based on gender. Female athletes and fans may be targeted with sexist insults and language meant to demean or belittle them based on their gender. Twenty-one out of 33 participants talked about unfair treatment on their social media platforms. If a mistake they made became public, they were more likely to receive abuse for it than men.

Participants described receiving comments suggesting that women do not belong in certain sports or that they should confine themselves to traditional gender roles.

“They tell me go to your house, your kitchen is your home, you have nothing to do with football, they always say it like that.” (interview 17)

All the comments women receive say that they do not belong, whereas when a mistake of a male sportsman becomes public, none of the comments he receives say that he does not belong in that sport.

One participant stated:

“If a female coach were to state a sports fact, many would not believe her. On the other hand, if a male coach were to state the exact same fact, everyone would believe him.” (interview 9)

Participants also stated that when men find a woman on a social media platform who seems to be better at a sport than they are, these men make comparisons and say things like “they are only better because they’re on steroids,” implying that no woman can be better than a man in any sport without additional help.

Theme 2: General misogyny

General misogyny in hate speech reflects underlying biases and prejudices against women in sports, specifically female athletes. There is a discrepancy in how female athletes and their performance are treated on social media: When they win, they do not receive the support and recognition they deserve. In contrast, when they lose, they are bombarded by negative comments. This suggests that the mistreatment is based on their gender rather than their actual performance.

Sixteen out of 33 participants mentioned that female athletes do not get any support from men if they win, but if they lose, they get many negative comments on their social media platforms.

“Yes, many tweets would be about the victory of the girls or about positive things if they had a victory or achievement, but I would not find any males in the comment section. But if there was a mistake or anything negative, males would be present in the comment section and they would criticize and say the reason for this is because she is a girl, if she was a boy, she would not have made that mistake.” (interview 14)

Some participants stated that when a women’s sports team loses, all the comments have comments such as that women do not belong in sports. However, when a men’s sports team loses, the comments may also be negative, but they never mention that men do not belong there.

“I mean okay, all of it is criticism, but they differ, I mean, for the girls, their criticisms are ridiculous, as if they say that you have no place in the field of sports and the men’s criticism, it has to do with football, criticism of the player, he is not playing well.” (10).

“In football matches, when a male makes mistakes, it’s normal, it’s a normal thing in the stadium, people make mistakes, but when they see it from a female, they see it as a lack of awareness of the thing they are doing, as if they are just playing and they don’t know anything in the first place, this is a societal view of women.” (interview 13)

Another way players can be affected is when their family names are out in public; most women are encouraged to participate under their first names only, so as not to focus attention on their families, which corresponds with the cultural values of the Middle East. This is closely tied to misogyny because it diminishes their recognition, reinforces traditional gender roles, and imposes conformity, reflecting sexist and misogynistic norms. Consequently, this practice may limit the personal recognition and visibility of these athletes, making it challenging for them to build individual brands and careers in sports.

“I mean I faced lack of support, but in terms of name, I mean, it’s OK to be public on social media, but the name shouldn’t come out, as in use a nickname.” (interview 10)

Theme 3: Appearance-related discrimination

Appearance-related remarks involve hate speech that centers around an individual’s physical appearance, including their body, clothing, or overall look. Eight participants said that they receive many negative comments if their physical appearance did not represent that of a typical woman. For instance, a participant stated: “I noticed they bully girls; they comment on their shapes and athletic bodies . Unfortunately, there is still a certain stereotypical thinking that girls should look a certain way, and this has nothing to do with it. Every person is free to appear in the way they like” (7).

Another participant talked about comments she has received: “Getting comments on the body in a very unbearable way;” “it can be sexual comments” (interview 32).

“ A strong girl who lifts heavy weights , for example, for sure she’s taking something, it’s impossible to be normal.” (27)

“I am going to be honest; boys see a girl who is muscled, sometimes they would come and say why is she more muscular than me and whatever else. I see it in the comments, they don’t like to see anyone better than them, I mean, they have the madness of greatness.” (interview 16)

“I see that with females, I mean they emphasize her outward appearance more than her skills…and if she makes a mistake they say, this is not your place; you should be studying or tending a home, or something like that, this is not for you.” (interview 6)

Although some excerpts touch on gender-related issues, they each highlight distinct forms of discrimination encountered by female athletes. For example, while it may seem that excerpt (interview 9) and (interview 27) are similar, excerpt (interview 9) sheds light on the credibility challenges female coaches encounter, such as receiving less recognition and support. In contrast, excerpt (interview 27) underscores the damaging stereotypes surrounding the physical appearance of female athletes. Thus, the excerpts are appropriately categorized under their respective themes based on the nature of the discrimination they illustrate.

Study 2

Methodology

As this research aims to develop an automated classifier to classify Arabic tweets using natural language processing (NLP) techniques for the purpose of detecting hate speech targeted at women in sports, the methodology covers data collation, annotation, pre-processing, classification, and evaluation. This approach is crucial for obtaining reliable and valid results in any research study. In this section, we outline the specific steps taken to achieve our research objectives, including the model architecture for Arabic hate-speech detection (Fig. 1).

Figure 1 Model Architecture for detecting hate speech in arabic tweets targeting women in sports.

The figure includes icons used for “evaluation”, “training”, “test set”, “train set”, data splitting”, “features extraction”, “pre-processing”, “data annotation” from Microsoft PowerPoint’s library and a Twitter icon directly from Twitter (https://twitter.com/).

Data collection.

To ensure a comprehensive dataset of Arabic tweets related to women in sports and hate speech, 7,487 tweets were collected from January 2021 to September 2023. The tweets were collected from X’s application programming interface (API) utilizing Tweepy library, which is a Python library to access the Twitter API. To ensure the relevance of tweets to our research topic, we leveraged keywords and phrases commonly associated with hateful comments, which were identified during our interviews with participants. Examples of these keywords include ” ” (women, sports, kitchen), ” ” (girls, sports, kitchen), ” ” (girls, sports, stupidity) and ” ” (women, sports, bodies). Additionally, we included relevant hashtags such as ” ” (Saudi women’s sports), ” ” (women’s sports), and ” ” (girls’ soccer) to ensure a comprehensive dataset reflecting various perspectives on women’s sports in Arabic. These keywords guided our search to collect tweets encompassing both hateful and non-hateful content.

Data annotation.

Annotation is crucial in developing a supervised machine learning model. In this case, the dataset was divided among six annotators (authors of this work), each responsible for labeling 1,247 tweets. All of annotators are Arabic-speaking Saudi female bachelor students majoring in computer science. The annotators were trained to identify tweets containing hate speech against women in sports. The qualitative data from interviews aided in the development of annotation criteria. Researchers used participant insights to identify what constitutes a hateful comment. The team encountered challenging cases during annotation, where the tweet’s content was ambiguous. To address these issues, meetings and discussions were held among the annotators to reach a consensus on label definitions, ensuring that all annotators were satisfied with the labelling criteria. This collaborative approach helps maintain the quality of the labeled data, which is essential for developing an effective machine learning model for detecting hate speech. To guarantee the precision and uniformity of the annotations, 1,247 tweets annotated by different annotators were randomly selected and reviewed by other annotators. The annotators agreed on most of the labels. The annotation process involved three distinct labels, as described in Table 1. This table provides the definitions, examples, and counts for each label used during annotation. To validate the team’s annotations, we hired an external annotator—an Arabic-speaking Egyptian male postgraduate in computer science—and paid $100 to review all tweet labeled as 0 and 1. The inter-annotator reliability between the team and the external reviewer was calculated using Cohen’s kappa (Cohen, 1960) (resulting in a strong agreement level of 91%.

Table 1 Overview of dataset labels.

Label	Definition	Count	Example	Translated example	
0(Support)	A tweet containing expressions that show support or encouragement for women participating in sports activities or events.	1,787		The best thing is that sports are beneficial for girls and their health. Congratulations to all Saudi girls.	
1(Hateful)	A tweet containing expressions that show negativity or hate towards women participating in sports activities or events.	2,025		Sports were ruined when women became its fans your place is in the kitchen.	
	The stupidity of girls continues and does not end cumulative genetic stupidity What did your mother know about football?	
	Women getting involved in sports is disgusting	
	God does not help who teaches girls football	
	Women’s bodies are built for KABSA and LUQAIMAT, what brought them to sports	
2(Irrelevant)	A tweet that does not contain any expression or opinion towards women participating in sports activities or events.	3,675		The account name should be Women’s Sports News.	

Data pre-processing.

Several cleaning and pre-processing procedures on the collected dataset were performed.

The initial step in cleaning the dataset involved the removal of duplicate tweets, aiming to reduce noise and minimize errors, resulting in a total of 7,487 tweets. Subsequently, the preprocessing pipeline focused on enhancing the quality of the text data. Firstly, both Arabic and English special characters, including hashtags, mentions, links, and punctuation marks, were eliminated to clean the text. Emojis present within the text were converted into corresponding words to. Following this, the tweets were tokenized, breaking them down into individual word units. Stop words were then removed to retain only meaningful words. Additionally, non-Arabic characters were filtered out to ensure the text’s exclusivity to Arabic. The normalization process further standardized the text by removing diacritics such as the double kasra tanween and , other variations. Additionally, it normalizes various forms of alif to and converts dotted yaa’ to . For example, following normalization, ” ” and ” ” are transformed into ” ” and ” ” Collectively, these preprocessing steps ensured that the Arabic tweets were thoroughly cleansed, standardized, and optimized for subsequent analysis and understanding.

Various Python libraries, version 3.11 using Anaconda 7, were utilized for the cleaning and pre-processing procedures, such as the Natural Language Toolkit (nltk) for tokenization and removing stop words, Regular Expressions (re) for pattern matching and string manipulations, String for string manipulation and punctuation removal, PyArabic.araby for Arabic-specific normalization functions, Scikit learn (sklearn) for encoding tweets label To further illustrate the cleaning and pre-processing steps, we present two examples of tweets before and after the prepossessing in Table 2

Feature extraction.

After performing essential data preprocessing tasks such as text cleaning and stop word removal, two distinct feature extraction methods were applied to the preprocessed tweets TF-IDF (Qaiser & Ali, 2018) and POS tagging. TF-IDF assigns weights to words based on their frequency within a document while considering their inverse frequency across the entire dataset. This approach helps identify significant words within each document, facilitating subsequent classification. The TF-IDF technique is commonly used in hate speech classification, because it effectively highlights important words in a text. Recent research, like that by Pariyani et al. (2021) and Mugambi (2017), has demonstrated the utility of TF-IDF for identifying hate speech on platforms like X and in other modern social media contexts. This technique is valued for its ability to uncover significant linguistic patterns in hate speech (Mugambi, 2017; Qaiser & Ali, 2018).

Table 2 Examples of tweets before and after applying pre-processing steps.

Before pre-processing	After pre-processing	
		

POS involves assigning a grammatical category (e.g., noun, verb, adjective) to each word in the text. This enriches the feature extraction process by providing information about the grammatical structure and contextual usage of words in the documents. The use of POS tagging in hate-speech classification has gained prominence due to its effectiveness in capturing linguistic nuances. Recent studies by (Watanabe, Bouazizi & Ohtsuki, 2018; Bauwelinck & Lefever, 2019) and Lefever emphasize the importance of linguistic analysis and emotional context, making POS tagging a relevant and valuable technique for hate-speech detection.

The tool we used to generate the features is the Arabic Linguistic Pipeline (ALP) developed by (Abed Alhakim et al., 2022). The choice for ALP was made based on a comparative study of the five most common Arabic POS taggers, the Stanford Arabic tagger (https://nlp.stanford.edu/software/tagger.shtml), CAMeL Tools (https://github.com/CAMeL-Lab/camel_tools), MADAMIRA (https://camel.abudhabi.nyu.edu/madamira), Farasa (https://farasa.qcri.org/POS), and ALP (http://arabicnlp.pro/). The study used text samples from Saudi novels and found that the ALP outperformed the other taggers (Alluhaibi et al., 2021).

ALP helped in the extraction of POS features and generated 245 different Arabic POS tags for class 0 and 225 tags for class 1, adding a valuable linguistic dimension to the feature set for classification. Table 3 shows a sample of tagged tweets classified as 1.

These two methods, TF-IDF and POS, were used independently, each contributing to the extraction of valuable features for classification.

Model training.

Following feature extraction, we trained several traditional supervised machine learning models to classify tweets as either hate speech or non-hate speech. While deep learning algorithms have shown promise in various NLP tasks, including hate speech detection (Badjatiya et al., 2017), they typically require very large amounts of data to perform well (Dhola & Mann, 2021; Mujahid et al., 2023) and our dataset size may not provide sufficient examples for effective training. Therefore, we chose traditional machine learning algorithms which are known to perform well with relatively small dataset size and can provide interpretable results.

In this study, four machine learning models from various classes have been applied, including linear, non-linear, tree-based, and non-tree-based methods. Most of these algorithms—SVM, Random Forest, Decision Tree, and XGBoost—are widely recognized for their effectiveness in hate speech text classification tasks across different languages. All models were implemented in Python version 3.11 using Anaconda 7. A brief description of each model is provided below:

Table 3 Sample of tagged tweets using the ALP tool.

Pre-Processed Tweets	Tagged Tweets using ALP	
	

• Support vector machines (SVM) (Cortes & Vapink, 1995): SVM is a widely used machine learning algorithm for classification and regression tasks. It works by maximizing the margin between the data points of different classes, using a decision boundary called a hyperplane.

• Random Forest (Breiman, 2001): Random Forest is an ensemble learning method that consists of multiple decision trees. It leverages a combination of bootstrap aggregating (bagging) and feature randomization to improve predictive accuracy. The individual decision trees are trained on random subsets of the data, and their outputs are combined to make predictions. This ensemble approach enhances model performance and reduces the risk of overfitting.

• Decision Tree (Quinlan, 1986): Decision Trees are hierarchical structures used for decision-making in classification and regression tasks. They recursively split the input data into subsets based on the values of input features, with the goal of maximizing the homogeneity of the resulting subsets.

• Extreme Gradient Boosting (XGBoost) (Chen & Guestrin, 2016): XGBoost is an optimized implementation of gradient boosted trees, which combines multiple weak classifiers (decision trees) to form a strong classifier. It uses a technique called boosting to iteratively improve the model by refining the weights of the weak classifiers.

These machine learning algorithms were selected due to their ability to handle high-dimensional data and their effectiveness in various text classification tasks. By training and comparing the performance of these models, we aimed to identify the best classifier for detecting hate speech against women in sports in the Arabic tweets dataset.

Results

Model evaluation

In our model evaluation, we conducted a thorough assessment of two distinct natural language processing tasks: TF-IDF and POS tagging. To validate our findings, we split the dataset into 70% for training and 30% for testing, and we employed a 10-fold cross-validation approach as suggested by Kohavi (1995). We measured the effectiveness of our trained models using standard machine learning performance metrics, including accuracy, precision, recall, and F1-score, which are widely recognized for evaluating model performance (Sokolova & Lapalme, 2009).

• True positives (TP): True positives represent the instances where the model correctly predicts positive cases.

• False positives (FP): False positives occur when the model incorrectly predicts positive cases that are actually negative.

• Accuracy: Accuracy is a measure of the proportion of correct predictions made by the model over the total number of predictions. It provides an overall assessment of how well the model is performing. Accuracy=TruePositives+TrueNegativesTruePositives+FalsePositives+TrueNegatives+FalseNegatives

• Precision: Precision is a metric that quantifies the accuracy of positive predictions made by the model. It measures the proportion of true positive predictions (correctly predicted positive cases) out of all positive predictions. Precision=TruePositivesTruePositives+FalsePositives

• Recall: Recall, also known as sensitivity or true positive rate, assesses the model’s ability to identify all relevant instances in the dataset. It measures the proportion of true positive predictions out of all actual positive cases. Recall=TruePositivesTruePositives+FalseNegatives

F1-score: The F1-score is a harmonic mean of precision and recall. It provides a balance between these two metrics and is particularly useful when you want to find a single metric that combines both precision and recall. F1score=2∗Precision∗RecallPrecision+Recallx.

The performance of the four machine learning models in classifying tweets as hateful or non-hateful was evaluated using two different feature sets: TF-IDF and POS features in Tables 4 and 5, respectively. In Table 4, utilizing TF-IDF features, all models demonstrated relatively strong performance. XGBoost stood out with the highest accuracy of 0.86 and precision and recall, each of 0.87 and 0.86, respectively. Random Forest also performed well, with an accuracy of 0.85 and precision and recall, each of 0.87 and 0.84, respectively. In Table 5, employing POS features, the models showed slightly lower but still reasonable performance. Random Forest had the highest accuracy of 0.84, with precision, recall, and F1-score values of 0.85, 0.83, and 0.83 respectively. SVM followed closely with an accuracy of 0.82 and similar precision, recall, and F1-score values of 0.84, 0.82, and 0.82 respectively. XGBoost showed also performance with an accuracy, precision, recall, and F1-score of 0.82. These results suggest that both TF-IDF and POS features are valuable in hate speech identification, and Random Forest consistently delivered good performance (0.85, 0.84) using TF-IDF and POS features respectively. Moreover, the results indicate that decision trees have the lowest performance, primarily attributed to their poor performance in text classification, as noted in Sun et al. (2011). This observation is supported by Pranckevičius & Marcinkevičius (2017), which consistently reports decision trees’ lower accuracy compared to models like Random Forest, logistic regression, and SVM in text reviews classification tasks.

Figures 2 and 3 illustrate the confusion matrix for the Random Forest model using TF-IDF and POS features respectively. The matrix is a visual summary of the classification accuracy shows the number of correct predications and number of errors and error types.

In conclusion, our evaluation of these machine learning models using various feature sets has indeed yielded valuable insights. For instance, we found that both the XGBoost and Random Forest models, when using TF-IDF features, excelled in detecting different forms of hateful text (achieving an accuracy of 86% and 85%, respectively), making them highly suitable for hateful speech classification tasks. Conversely, the Random Forest model demonstrated the best accuracy (84%) compared to others when utilizing POS features, establishing it as a strong choice for hateful speech classification tasks. These findings provide crucial guidance for selecting the appropriate machine learning models for specific NLP tasks, ensuring the optimal model is chosen for the job.

Table 4 Performance metrics for different machine learning models using TF-IDF featuers.

Model	Accuracy	Precision	Recall	F1-Score	
SVM	0.85	0.86	0.85	0.85	
Random Forest	0.85	0.87	0.84	0.84	
Decision Tree	0.81	0.81	0.81	0.81	
XGBoost	0.86	0.87	0.86	0.86	

Table 5 Performance metrics for different machine learning models using POS featuers.

Model	Accuracy	Precision	Recall	F1-Score	
SVM	0.82	0.84	0.82	0.82	
Random Forest	0.84	0.85	0.83	0.83	
Decision Tree	0.77	0.77	0.77	0.77	
XGBoost	0.82	0.82	0.82	0.82	

Figure 2 Confusion matrix for the random forest model using TF-IDF features.

Figure 3 Confusion matrix for the random forest model using POS features.

Important features

Finding and showing the most important features in a dataset used for binary classification is crucial. It helps pick the best features and provide insights into which feature(s) affect the model’s predictions the most. Figure 4 displays the top five POS features from a pool of 263 features, utilizing the Random Forest classifier. Below, a brief overview of each color-coded feature depicted in Fig. 4:

Figure 4 Important features by class.

“Definite article + PIN: Plural irregular noun”: Refers to a plural noun with the definite article and inflectional endings.

“D: Definite article + SFAJ: Singular feminine adjective”: Describes a singular feminine adjective preceded by the definite article.

“D: Definite article + SFN: Singular feminine noun”: Indicates a singular feminine noun with the definite article.

“SFN: Singular feminine noun”: Simply a singular feminine noun without a definite article.

“SMN: Singular masculine noun”: Denotes a singular masculine noun without the definite article.

These linguistic features provide essential information about the grammatical attributes of Arabic words, aiding in the analysis and comprehension of Arabic texts by revealing details like gender, number, and definiteness. For instance, the label “D: + SFAJ ” indicates a singular feminine adjective preceded by the definite article, enabling readers to identify the precise grammatical function of the word in context.

These linguistic features shed light on how hate speech towards women in sports is linguistically constructed. Plural irregular nouns with the definite article may imply collective derogatory references to female athletes. Meanwhile, the presence of definite articles with singular feminine adjectives and nouns suggests individualized attacks on specific female figures. Conversely, the absence of the definite article in singular feminine and masculine nouns may indicate a shift towards generalized or dehumanizing language targeting women or men in sports. These features offer valuable insights into the linguistic dynamics of hate speech in the sports domain.

A potential integration of qualitative findings with linguistic analysis may allow exploration of how specific linguistic features, such as the presence or absence of definite articles in plural and singular nouns, correlate with various hate speech themes directed at women in sports. For instance, the presence of plural irregular nouns with the definite article may align with gender-based derogatory comments, while the absence of the definite article in singular masculine nouns might relate to instances of general misogyny.

It’s crucial to recognize the complexity of Arabic part-of-speech (POS) tagging, arising from its rich morphology. The intricate structure, with numerous inflections, poses challenges in transferring POS features to other languages. Arabic’s effectiveness in POS tagging relies on understanding its unique morphological and syntactic features, which may not easily translate to languages with different structures. Despite these challenges, exploring the applicability of these features in other languages is important.

Discussion

This research provides a substantial contribution to the expanding body of literature on identifying hate speech on social media, with a specific focus on women in sports and the Arabic language. The research aims to address two key research questions. Firstly, RQ1 focuses on understanding the various forms of online hate speech experienced by female athletes and women’s sports in Arabic. The qualitative study reveals prevalent themes such as gender-based derogatory comments, general misogyny, and appearance-related discrimination. While some initially viewed hateful speech as motivation, negative comments eventually took a toll on their emotional well-being, emphasizing the need for further exploration of its effects.

Secondly, RQ2 involves the development and evaluation of machine learning models to identify Arabic hateful tweets targeting women in sports. To achieve this, a unique collection of Arabic tweets related to women in sports was categorized as hateful or non-hateful. These tweets were manually labeled for reliability. Insights from qualitative findings guided the creation and annotation of datasets, with common keywords and phrases associated with hateful comments informing the annotation criteria. Two types of feature extraction techniques (TF-IDF and POS tagging) were applied to the corpus dataset. Experiments with four machine learning algorithms demonstrated the effectiveness of the XGBoost model, combined with TF-IDF features, in hate speech detection. Additionally, the Random Forest model, coupled with POS tagging features, proved efficient due to its ensemble learning approach and linguistic insights provided by POS tagging.

To sum up, the current research is not merely an exploration of the problem, but a significant step towards making a tangible impact on the wellbeing and performance of female athletes in the digital age. The research introduces a ML model to identify Arabic cyberhate speech targeting female sports that can be leveraged as a crucial step toward automating the detection of such content, laying the foundation for the development of automated systems capable of proactively flagging and addressing hateful tweets in real time.

Limitations and Future Research Directions

While this study focuses on online hate against female athletes in the Arabic world, we recognize its broader relevance. Our findings extend to discussions on strategies for combating online hate speech across diverse cultural and linguistic contexts. Also, the methodology employed in the study can be a framework for similar studies, enabling comparative analyses and pattern identification. Also, it is essential to acknowledge that the findings of study 1 were derived from interviews with a sample of only 33 participants. While these interviews provided valuable insights, expanding the sample size in future research is important to gain a more comprehensive understanding of the challenge’s women in sports face in the digital realm.

Additionally, the relatively small corpus size may also constrain the applicability of the findings. This limitation, in part, arises from X’s active removal of hate speech upon user reporting, rendering it challenging to amass a larger dataset. Expanding the dataset and including a broader set of features could lead to more robust and accurate models for detecting hate speech against women in sports. Additionally, future research may delve into advanced machine learning algorithms (e.g., deep learning), with due consideration for the dataset’s size limitation.

Conclusions

To the best of our knowledge, the dataset constructed for this research is the first annotated Arabic dataset of hate speech against women with a special focus on the sport domain. The dataset can be extended for future investigations, aiming to discern the distinctions between discriminatory and gendered hate language directed at women in sports and hate language in Arabic used across other domains (e.g., politics, religion). By employing varied approaches in dataset creation, researchers can explore whether these different contexts share common characteristics that could augment the effectiveness of hate speech detection. Although laws and regulations by authorities such as governmental organizations and human rights organizations are exerting pressure on social media companies to properly address the phenomenon of hate speech on their platforms, the majority of these companies depend upon users to flag such information, which is subsequently subject to some kind of manual screening, and this strategy appears to be impractical (Ombui, Muchemi & Wagacha, 2019). This study sheds light on the role that automated solutions can play in resolving the problem.

Hate speech against women in sport not only humiliates sportswomen, degrades their participation in sports and leads to reduced worker productivity, but also has an impact on expenditures in the health sector and the sports industry. Key stakeholders (e.g., Ministry of Sport, social media platforms) should take additional steps to address the challenges of combating this, for example through educational programs, regulations, and awareness initiatives to foster an inclusive and safe sports environment for women, promoting their participation free from discrimination.

Supplemental Information

Supplemental Information 1 Participant Demographics

Supplemental Information 2 Interview Questions

Supplemental Information 3 Python script implements a machine learning classification model using TF-IDF features

Supplemental Information 4 Python script implements a machine learning classification model using POS features

Supplemental Information 5 Python script for identifying most important POS features in text classification

Supplemental Information 6 Questionnaire (Original Language, Arabic)

We extend our gratitude to Dr. Hanen Himdi for her invaluable assistance in conducting the analysis related to POS tagging.

Additional Information and Declarations

Competing Interests

Author Contributions

Ethics

Data Availability

The authors declare there are no competing interests.

Fatimah Alhayan conceived and designed the experiments, performed the experiments, analyzed the data, performed the computation work, prepared figures and/or tables, authored or reviewed drafts of the article, and approved the final draft.

Monerah Almobarak conceived and designed the experiments, performed the experiments, analyzed the data, performed the computation work, prepared figures and/or tables, and approved the final draft.

Hawazen Shalabi performed the experiments, analyzed the data, authored or reviewed drafts of the article, and approved the final draft.

Luluwah Alshubaili conceived and designed the experiments, performed the experiments, analyzed the data, authored or reviewed drafts of the article, and approved the final draft.

Renad Albatati analyzed the data, authored or reviewed drafts of the article, and approved the final draft.

Wafa Alqahtani analyzed the data, authored or reviewed drafts of the article, and approved the final draft.

Nofe Alhaidari performed the experiments, performed the computation work, prepared figures and/or tables, and approved the final draft.

The following information was supplied relating to ethical approvals (i.e., approving body and any reference numbers):

Ethical approval was obtained from the Institutional Review Board of PNU University before commencing the study (23-0734)

The following information was supplied regarding data availability:

The raw data are available at GitHub: https://github.com/monerahalmobarak/Automatic-Detection-of-Cyber-Hate-Speech-Towards-Female-Sports-in-the-Arabic-X-Sphere.

The dataset contains public tweets that were collected and labeled as ‘0’, ‘1’, or ‘2’ based on their classification criteria. These tweets are used for the analysis in Study 2.

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
