# Peer review of "Detection of cyberhate speech towards female sport in the Arabic Xsphere"

_PeerJ Computer Science, doi:10.7717/peerj-cs.2138_

## Round 0.1 · original submission · Major Revisions

All three reviewers appreciated the importance of the topic and generally, the approach taken in the paper. While reviewer 2 suggested 'Minor revisions", and the other two reviewers - "Major revisions", all three reviewers found that important parts of the experimental design that are missing and need to be provided in the revised version: a justification for the machine learning algorithms used, the raw data is not shared, details to help contextualize the qualitative findings, reference to the source of the interview questions used in the study. The discussion needs to be improved to make clear how the research questions are answered by the study. Also, the quality of the figures needs to be improved, an English translation should be provided for figures which contain Arabic and French. All three reviewers have provided valuable suggestions for improvement, which should be considered. The size of the revisions necessary is significant, therefore the recommendation is "Major revision".

Reviewer 1 ·

Basic reporting

The topic of the research is really interesting, but the question that comes to mind is how the authors can generalize their results and how they can generalize the topic to attract readers' attention. The topic is specific to female sports in the Arabic world.

The paper is too long, and it is suggested to summarize some parts of it, such as the introduction, and focus more on defining the problem and emphasizing its importance.

Is there a specific reason why the accuracy of decision trees is lower than other methods, and how can this be justified?

Why were only four machine learning models used? What about the other models, and why weren't they considered for use?

In the important features section, authors expressed in Arabic language, and it seems better explained in English. Additionally, it's important to explore how these features are applicable in other languages similar to Arabic, for example, French.

In the discussion, RQ1 and RQ2 should be discussed in more detail. Readers are unable to find the answers to the research questions based on the requirements of this study.

The quality of the figures and tables is poor. They need to be replaced with new versions.

Experimental design

As above

Validity of the findings

As above

Cite this review as

Reviewer 2 ·

Basic reporting

I thank the authors for their easy to read paper; it is well-written; the vocabulary that they used is easy to read and understand for international readers. In addition, I like how authors divided the introduction into sections based on different aspects that the paper discusses. That’s make it easier to understand the problem and the literature behind it. The paper also includes sufficient introduction and background information.

However, there is some typos

• In line 16: “recent” written as “resent”
• Randomforest (without space ) instead of Random Forest in line 624,628, 630
• TFIDF line 33 , it is usually written as TF/IDF or TF-IDF, I recommend to use a separator as found in literature and write it in a consistent way throughout the text.
• Figure 5, I think it is class 0 and class 1, not 2
• In line 623, in table 5 not table 4 (table 5 is the one that shows the results of POS)
• Its better to provide the formulas (equations) of the performance metrics (precision, recall, and F1-score), it will be easier to read and understand than only text description.
• Section 1.3 discusses the sports and its benefits but in a general. There is no clear connection between this section and other sections. At the beginning, I thought authors will discuss the hate speech in sport despite the gender, but I realized that it is not. I recommend removing this section.
• I thank the authors for providing the abbreviation meanings of some abbreviations; however, they did not provide the meanings of others, such as, SVM in line 230, LSTM in line 242, GBDT in line 245, GloVe in line 243. I recommend providing the meaning of any abbreviation used in the paper.
• Despite that authors provide good references for most of the evidences they gave, there are missing references for some information, for example:
o Line 48:” significant growth in the number of users has made social media a crucial aspect of modern life”
o Line 59: “Various studies support the dire consequences of the prevalence of hate speech”
o Line 226-227: “Hate speech is generally detected using classical machine learning; only recently has it begun to be detected with deep learning.”
o Line 269-270 : “Hate speech on social media targeted at women in sports is rapidly increasing. Furthermore, it has recently been a controversial topic in Saudi Arabia.”
o Line 270-271:” few studies have been investigated the issue of women sport on Arabic language”
o Line 334: “The interview questions (see Appendix A) were designed drawing from prior research and based on methodological considerations”
• They provide the new name of Twitter, “X”; I recommend that they mention the new name in the abstract since it is first mentioned in the abstract. I recommend using X instead of Twitter afterwards.
• For increasing the readability, I prefer to write “interview #” before the interview number in line 367, line398,408,413,423,433,436,439,444,and line 448

Experimental design

• I thank the authors for providing enough explanations of the data analysis methods in both studies; they also explained how they link study 1 with study 2 clearly. They explain in details and provide good references of all methods used in study 2, how they collect the data, how they made data annotation, data pre-processing, feature extraction, and model training.
• Authors did not provide the references that they draw the interview questions from
• The raw data is not provided
• In study 1: they raise RQ1: What is the extent of online hate speech experienced by female athletes and female sports fans on social media, and what are the specific forms it takes?
• However, they answered part 2 of the question, which is the forms of hate speech experienced by female athletes and female sports fans on social media. I recommend removing the first part “What is the extent of online hate speech experienced by female athletes and female sports fans on social media”
• In line 465 through line 468, authors mentioned some examples of relevant keywords and hashtags that they used in their search. The terms that they provide all neutral, none of these examples is hateful terms. It is very important to provide the list of all terms and hashtags that are used in search. They can make this list in a table.
• In line 503 through line 505, authors gave example of how they make some pre-processing through stemming, however in table 2, the example did not show this idea, "الرياضة" after pre-processing should converted to “رياض”; for this reason, I recommend providing better examples.
• I recommend adding examples of class 0 in table 3 since table 3 now contains examples of class 1 only

Validity of the findings

The results are clear and they a well-known performance metrics: accuracy, precision, recall and F1-score. They showed clearly the results in table 4 and table 5. However, table 5 showed that the best results is for Random Forest (0.84,0.85,0.83,0.83) followed by SVM (0.82,0.84,0.82,0.82) not XGBoost (0.82,0.82,0.82,0.82) which conflict with the results mentioned in the text in line 625 and 626.

Additional comments

all of my comments are to improve the manuscript

Cite this review as

·

Basic reporting

The manuscript is well-prepared and informative. However, there are some issues with in-text citation styles. For example, in Lines 180, “(Kearns et al., 2022) provide a scoping overview of … “. Should it be “Kearns et al. (2022) provide a scoping overview of …”? Several items in the References are not complete. For example, the item listed in Lines 786-789 does not have complete journal title.
Figure 1. Common forms of hate speech targeting women in sports may not be needed as the same categories are detailed in the corresponding section.
Another suggestion is that the authors should provide English translation for the Arabic content in the manuscript, for example, the Arabic tweet in Table 1.

Experimental design

It is commendable that the authors combined the qualitative approach and quantitative approach in this study to better understand the hate speech faced by female athletes. However, some more details may be needed to help contextualize the qualitative findings. For example, nationality of the interviewees, and number of participants in each subgroup (athletes, sports fans). In addition, the authors may want to provide a meaningful research id (to indicate participant demographic info) for each of the excerpts so that readers can better make sense of their comments.
In Study 2, the authors may want to share a bit more about the annotators, for example, basic demographic information and academic background, as well as the inter-annotator reliability information.
The authors may want to share the tools they used for model training. Also, since they decided not to use deep learning models, which are mentioned in the literature review section, they may want to mention their decision and rationale earlier.

Validity of the findings

About the qualitative results, the first two themes (‘gender-related derogatory comments’ and ‘general misogyny’) seem to have some excerpts that share certain features, for example, sense of belonging as expressed in (17) and (10) in section 3.1.3. Likewise, Excerpt from (27) in Theme 3 seems to be similar to the excerpt (9) in Theme 1
The authors claim that “the qualitative findings served as the foundation for creating and annotating datasets to address the second research question”. This connection is not very clear at least in the description of Study 2. For example, what ‘keywords and phrases commonly associated with hateful comments’ are used from the interview data? What criteria for detecting gender-related hate speech were derived from the interview data?
Also, the qualitative findings do not seem to contribute to the feature extraction efforts at all.
In 3.3.2 Important Features, the authors discussed five most important POS features. Those features look like POS n-grams, rather than individual POS features, to me. In the discussion of these features, it would be interesting to see how these features are related to the themes (and/or examples) identified in the qualitative analysis.

---

## Round 0.2 · accepted · Accept

Thank you for adequately addressing the reviewers' comments. Two of the three reviewers recommended accepting your paper, and I agree with their decision.

Reviewer 2 ·

Basic reporting

I thank the authors for their easy to read paper; it is well-written; the vocabulary that they used is easy to read and understand for international readers. In addition, I like how authors divided the introduction into sections based on different aspects that the paper discusses. That’s make it easier to understand the problem and the literature behind it. The paper also includes sufficient introduction and background information.

They have made all the corrections and modifications that mentioned in all reviewers' comments of previous version.

Except:
they forget adding "reviewer" word before the number of reviewer number in line 379, 404, 410

the first appearance of Twitter was in line 19 (in the abstract) so it is better to change to X (formerly known as Twitter)

Experimental design

I thank the authors for providing enough explanations of the data analysis methods in both studies; they also explained how they link study 1 with study 2 clearly. They explain in details and provide good references of all methods used in study 2, how they collect the data, how they made data annotation, data pre-processing, feature extraction, and model training.

They have made all the corrections and modifications that mentioned in all reviewers' comments of previous version.

Validity of the findings

They have made all the corrections and modifications that mentioned in all reviewers' comments of previous version.

Additional comments

Thanks alot for the authors for the efforts to make the manuscript better

Cite this review as

·

Basic reporting

It appears that the new version has been carefully edited.

Experimental design

No further comment on this aspect.

Validity of the findings

No further comment on this aspect.

Additional comments

With newly added information and further clarification of key concepts, thiss revised version has been greatly improved. The authors have addressed reviewers' comments adequately. I believe it is now in good shape for publication.